Element concentrations in pelagic Sargassum along the Mexican Caribbean coast in 2018-2019

Rodríguez-Martínez Rosa E. rosaer@cmarl.unam.mx 1
Roy Priyadarsi D. 2
Torrescano-Valle Nuria 3
Cabanillas-Terán Nancy 3 4
Carrillo-Domínguez Silvia 5
Collado-Vides Ligia 6
García-Sánchez Marta 1 7
van Tussenbroek Brigitta I. 1
1 Instituto de Ciencias del Mar y Limnología, Universidad Nacional Autónoma de México , Puerto Morelos , Quintana Roo , México
2 Instituto de Geología, Universidad Nacional Autónoma de México , Ciudad de México , Ciudad de México , México
3 El Colegio de la Frontera Sur, Unidad Chetumal , Chetumal , Quintana Roo , México
4 CONACYT - El Colegio de la Frontera Sur , Chetumal , Quintana Roo , México
5 Departamento de Nutrición Animal, Instituto Nacional de Ciencias Médicas y Nutrición “Salvador Zubirán” , Ciudad de México , Ciudad de México , México
6 Department of Biological Sciences and Center for Coastal Oceans Research in the Institute of Environment, Florida International University , Miami , FL , United States of America
7 Instituto de Ingeniería, Universidad Nacional Autónoma de México , Ciudad de México , Ciudad de México , México
Anderson Todd
Electronic publication date: 2020 Feb 26
Publication date: 2020
Volume: 8
Electronic Location ID: e8667
Received 2019 Nov 25; Accepted 2020 Jan 30
Copyright: ©2020 Rodríguez-Martínez et al.
Copyright year: 2020
Copyright holder: Rodríguez-Martínez et al.
License: This is an open access article distributed under the terms of the Creative Commons Attribution License, which permits unrestricted use, distribution, reproduction and adaptation in any medium and for any purpose provided that it is properly attributed. For attribution, the original author(s), title, publication source (PeerJ) and either DOI or URL of the article must be cited.
License URL: https://creativecommons.org/licenses/by/4.0/

Keywords: Sargassum fluitans, S. natans, Metal content, Arsenic, Caribbean Sea

Funding: The authors received no funding for this work.

==============================
The massive influx of pelagic Sargassum spp. (sargasso) into the Mexican Caribbean Sea has caused major deterioration of the coastal environment and has affected the tourism industry as well as livelihoods since 2015. Species of Sargassum have high capacity to absorb metals; thus, leachates of sargasso may contribute to contamination by potentially toxic metals when they drain into the sea and into the groundwater when dumped in inadequate land deposits. Valorization of sargasso would contribute to sustainable management; therefore, knowledge on potentially toxic metal content is necessary to define possible uses of the algae. We present concentrations of 28 elements measured using a non-destructive X-ray fluorescence analyzer (XRF) in 63 samples of sargasso collected between August 2018 and June 2019 from eight localities along ∼370 km long coastline of the Mexican Caribbean Sea. The sargasso tissues contained detectable concentrations of Al, As, Ca, Cl, Cu, Fe, K, Mg, Mn, Mo, P, Pb, Rb, S, Si, Sr, Th, U, V, and Zn. The element concentration in sargasso varied on spatial and temporal scales, which likely depended on the previous trajectory of the pelagic masses, and whether these had (or had not) passed through contaminated areas. Total arsenic concentration varied between 24–172 ppm DW, exceeding the maximum limit for seaweed intended as animal fooder (40 ppm DW) in 86% of the samples. For valorization, we recommend analyses of metal contents as a mandatory practice or avoiding uses for nutritional purposes. The high arsenic content is also of concern for environmental contamination of the sea and aquifer.

Introduction

The west coast of Africa and some eastern Caribbean islands received unusual large quantities of pelagic Sargassum spp. (S. fluitans (Boergesen) Boergesen and S. natans (Linnaeus) Gallion; hereafter named sargasso) for the first time in 2011 (Gower, Young & King, 2013). In subsequent years, the range of massive sargasso influx extended over the Atlantic Ocean and whole Caribbean Sea. Wang et al. (2019) reported more than 20 million metric tons of sargasso in the open ocean in the peak month of June 2018, when the Great Atlantic Sargasso Belt extended for 8,850 km in total length. Beaching of sargasso has caused havoc to the Caribbean coastal ecosystems. Leachates and particulate organic matter from stranded decaying algal masses depleted the oxygen in near shore waters and reduced visibility of the water column, causing mortality of near-shore seagrasses and fauna (van Tussenbroek et al., 2017; Rodríguez-Martínez et al., 2019). Onshore and near shore masses of sargasso interfered with the seaward journeys of the juvenile turtles (Maurer, De Neef & Stapleton, 2015), affected sea turtle nestings (Maurer, Stapleton & Layman, 2018) and altered the trophic structure of the sea urchin Diadema antillarum in coastal marine systems (Cabanillas-Terán et al., 2019). Massive beachings also enhanced beach erosion (van Tussenbroek et al., 2017). Coastal ecosystem-based tourist industry is one of the major sources of income for the Caribbean countries (Langin, 2018) and the potential socio-economic impacts of ecosystem degradation due to sargasso influx have yet to be assessed.

The Mexican Caribbean coast began receiving massive amounts of sargasso during the late 2014 and it reached a peak in September 2015, when in the northern section of the coast between Cancun and Puerto Morelos an average of ∼2,360 m3 of algae (mixed with sand, seagrasses and other algae) arrived per km of coastline (Rodríguez-Martínez, van Tussenbroek & Jordán-Dahlgren, 2016). During 2016, and 2017, the influxes decreased, increasing again in 2018, when in the peak month May ∼8,793 m3 km−1 of algae (mixed with sand, seagrasses and other algae) were removed from the same shore section (Rodríguez-Martínez et al., 2019). In the tourist beaches, the algae removed from the beach and sea have been disposed in areas that are not properly prepared to avoid leakage of the leachates into the aquifer. In addition, the cleaning efforts have not covered the whole coastline and thousands of tons of sargasso have accumulated annually along the Mexican Caribbean coast.

Like other brown algae, species of Sargassum (including the pelagic ones) have high capacity to absorb metals and other elements (Kuyucak & Volesky, 1988; Davis, Volesky & Vieira, 2000). This high absorption capacity is attributed to the unique mixture of polysaccharides, mainly alginates, in their cell walls (Fourest & Volesky, 1997). At present, the Sargassum spp. are used for different commercial end products, such as fertilizers (Milledge & Harvey, 2016), textiles, paper and drugs (Oyesiku & Egunyomi, 2014), as well as in the production of biogas (Wang et al., 2018). They have also been increasingly used as food for animals and humans, and therefore the high concentrations of contaminants, including heavy metals, may pose potential health risks (Reis & Duarte, 2018). Therefore, it is mandatory to evaluate elemental concentrations to ensure that acceptable levels are maintained in terms of health regulations (e.g., Fourest & Volesky, 1997). Previous studies on metal contents in sargasso, were either based on limited number of samples collected mostly from a single locality (e.g., Nigeria Oyesiku & Egunyomi, 2014; Dominican Republic, Fernández et al., 2017) or in a single season (e.g., Addico & De Graft-Johnson, 2016). Hence, it is unclear how much the metal contents can vary in the algal tissues across sites and seasons and between species.

In this study, we estimate concentrations of 28 different elements in sargasso tissues collected from the Mexican Caribbean coast, covering a linear north-south distance of 370 km. We hypothesize that the elemental contents are variable both in time and space. The determinations of metals and other elements from this study provide an essential baseline data for adequate management and potential uses of sargasso.

Survey Methodology

Study sites

We collected 63 samples of sargasso along the Mexican Caribbean coast, from Contoy Island, at the northern extreme, to Xcalak in the south (Fig. 1). This region receives an average precipitation of ∼1,061 mm y−1 and the sea-surface temperature (SST) ranges from 25.1–29.9 °C (Rodríguez-Martínez et al., 2010). The Yucatan Current, a major branch of the Caribbean Current, transports the pelagic algal masses parallel to the Mexican Caribbean coastline. Easterly trade-winds dominate this region during the summer and mild cold fronts occur during the winter season. Trade-winds transport the superficial waters towards the shore, importing the pelagic masses of sargasso towards the coast.

Figure 1 Sampling sites.

Location of the sampling sites of sargasso along the Mexican Caribbean coast between August 2018 and June 2019. Map produced in QGIS 2.18 (http://www.qgis.org) using the following data sources: National Geospatial-Intelligence Agency (base map, World Vector Shoreline Plus, 2004. http://shoreline.noaa.gov/data/datasheets/wvs.html). The location of survey sites was obtained from the present study. Data sources are open access under the Creative Commons License (CC BY 4.0).

The coastal environment consists of beaches, rocky shores, seagrass beds, coral reefs, mangroves, jungle and underground rivers (Hernández-Arana et al., 2015). All these ecosystems provide services to the tourism industry, a crucial component of the regional economy (Spalding et al., 2017). In the karstic Yucatan peninsula, the freshwater aquifer and seawater are constantly interacting; especially near the coast (Hernández-Terrones et al., 2011; Hernández-Terrones et al., 2015). This region has no other major industries besides tourism. At present, this region has the highest number of hotel rooms in Mexico and the number of rooms has increased from 3,206 in 1975 to 100,986 in 2017 (SEDETUR, 2019). Similarly, the resident population grew almost 15-folds, from less than 100,000 in 1970 to 1,501,785 in 2015 (INEGI, 2015). This rapid urban development has caused coastal pollution through influx of nutrients (Carruthers, van Tussenbroek & Dennison, 2005; Hernández-Terrones et al., 2011; Baker, Rodríguez-Martínez & Fogel, 2013; van Tussenbroek et al., 2017), sewage (Metcalfe et al., 2011), and some metals (e.g., Lead, see Whelan III, van Tussenbroek & Santos, 2011) into the coastal ecosystems.

Methodology

Field collection

Samples were collected between August 2018 and June 2019 from eight different sites along the Mexican Caribbean coast (from north to south): (1) Contoy Island, (2) Blue waters, (3) Puerto Morelos, (4) Cozumel, (5) Mahahual, (6) Chinchorro, (7) Xahuayxol and 8) Xcalak (Fig. 1, Table 1). Fresh sargasso (golden color) thalli floating near the shore (2–20 m) and in the ocean (>5 km from shore) were collected manually and separated in species and morphotypes (S. fluitans III, S. natans I and S. natans VIII) in the laboratory following Schell, Goodwin & Siuda (2015), except for the samples of Contoy Island (CI). The samples collected from CI were frozen before separating the specimens by species and morphotypes, thus, we classified them as Sargassum spp. All the samples were placed in an oven for at least 48 h at 60 °C until completely dry. Special caution was taken to avoid contact between the algal samples and any metal object. Samples were shipped to the Institute of Geology of the National Autonomous University of Mexico for the analysis of element concentrations. We did not remove epibionts from the thalli and analyzed the chemical composition of the algae including attached organisms, as the main interest of this study was to determine the potential contamination hazards and uses of sargasso as collected from the sea, without any specific separation treatment. All surveys were conducted under permit PPD/DGOPA-116/14 granted by SAGARPA (Agriculture, Natural Resources and Fisheries Secretariat) to B.I. van Tussenbroek.

Table 1 Samples information.

Number of samples collected at eight sites along the Mexican Caribbean coast during 2018–2019. Habitat refers to distance from coast, shore (2–20 m from coast) or ocean (>5 km from coast).

Locality	Habitat	Collection	Species/Morphotype	Total	
		Year	Month	Sflu III	Snat I	Snat VIII	Sarg sp		
1 - Contoy Island	Ocean	2019	March				4	4	
2 - Blue waters	Ocean	2018	August		1			1	
3 - Puerto Morelos	Shore	2018	August	1	1	1		3	
			September	1	1	1		3	
			October	2	1	2		5	
			December	1	1			2	
		2019	February	2	2	2		6	
			March	1	3	2		6	
			April	1		1		2	
4 - Cozumel	Ocean	2018	August			1		1	
		2019	May	1	1	1		3	
5 - Mahahual	Shore	2019	May	4	3			7	
6 - Chinchorro	Shore	2019	May		1			1	
7 - Xahuayxol	Shore	2019	April	1	2				
			May	2	1				
			June	3	3				
8 - Xcalak	Shore	2019	May	3	3			6	
Total				24	24	11	4	63	
Notes.

Sarg sp: Sargassum spp., Sflu III: Sargassum fluitans III, Snat I: S. natans I, Snat VIII: S. natans VIII.

Elemental analysis

Concentrations of 28 different elements were measured in dry samples using a Niton FXL 950 energy dispersive X-ray fluorescence (XRF) containing a 50 kV X-ray tube of Ag and equipped with a geometrically optimized large area drift defector following Quiroz-Jiménez & Roy (2017). Table S1 shows the limit of detection of these elements. The dried samples were processed in the laboratory using a non-destructive sample preparation technique. Approximately 5–7 dry g of each sample was placed in a plastic capsule that has a 4µm thick polypropylene X-ray film on one side and the other side of the capsule was packed with synthetic flexible gauze. The samples were measured in the mining Cu/Zn mode and three different filters using the internal calibration curves previously generated by comparing the results of Niton FXL with a conventional XRF (e.g., Quiroz-Jiménez & Roy, 2017). The results are expressed in parts per million dry weight (ppm DW) after carrying out the analysis in five repetitions in each sample. We used two different geological reference materials (Es-2, organic rich argillite and Es-4, dolostone) for estimation of precision (Kiipli et al., 2000). Except for Mg, all other elements have relative standard deviation (RSD) between <1 and 5%. Mg concentrations show RSD of 26% and it is the least precise among all the analyzed elements. Some advantages of the XRF analysis compared to other methodologies are that small samples are required (∼5 g), the results have high precision, and it is non-destructive, permitting the same sample to be reused for other studies. Also, it is less expensive and faster compared to the use of an ICP-MS. The relatively high limit of detection of XRF for some elements is a disadvantage, and some potentially toxic elements may have been present in low concentrations, but were not measured (e.g., Ni and Co). This technique measures concentrations independent of the chemical state of an element.

Table 2 Element concentrations median and range.

Element concentrations (ppm DW) of pelagic Sargassum spp. tissue collected from eight localities along the Mexican Caribbean coast between 2018 and 2019. The number of samples with readings above LOD are expressed in % of the total sample size (n = 63).

Element	LOD	Samples with readings above LOD (%)	Minimum	Maximum	Median	
Al	140	58.7	<LOD	500	206	
As	4	100	24	172	80	
Ca	394	100	23,723	136,146	70,040	
Cl	266	100	747	53,101	22,350	
Cu	6	7.9	<LOD	540	<LOD	
Fe	3	7.9	<LOD	11	<LOD	
K	333	100	1,990	46,002	19,666	
Mg	2,915	92.1	<LOD	13,662	6,537	
Mn	13	100	40	139	71	
Mo	1	7.9	<LOD	7	<LOD	
P	145	100	228	401	327	
Pb	2	7.9	<LOD	3	<LOD	
Rb	1	100	30	143	56	
S	199	100	9,462	24,773	14,363	
Si	342	100	447	2,922	1,767	
Sr	6	100	1,605	2,564	1,890	
Th	1	100	5	23	10	
U	4	100	11	48	23	
V	3	28.6	<LOD	13	<LOD	
Zn	5	12.7	<LOD	17	<LOD	
Notes.

LOD, Limit of detection.

Data analyses

The median of the five readings per element of each sample was calculated and used for further analysis. For each element, the readings below the limit of detection (<LOD; Table S1) were substituted with LOD/ 2 for calculation of summary statistics (Celo & Dabek-Zlotorzynska, 2010). Distributions (spread of data and the median values) of the fourteen most commonly found elements (e.g., Al, As, Ca, Cl, K, Mg, Mn, P, Rb, S, Si, Sr, Th and U) in sargasso tissue for each sampling locality are illustrated by dot plots. Differences in the concentration of elements among species and morphotypes were tested using non-parametric ANOVAs based on the Kruskal–Wallis rank procedure. We constructed a heatmap using the data from fourteen elements from Puerto Morelos (location 3, see Fig. 1) to visualize temporal differences in concentration of metals in seven different sampling periods between August 2018 and April 2019. Element concentration values were Z-score-transformed across sampling times and their values above and below the mean were used to generate the heatmap. The Z-value is a dimensionless quantity which is defined by the following equation (Larsen & Marx, 1986): Z=X−μ∕σ

Where X represents an individual raw score that is to be standardized, σ is the standard deviation of the population, and μ is the mean of the population.

All analyses were done in R (R Core Team, 2019) using packages: dplyr (Wickham et al., 2019), ggplot2 (Wickham, 2009), gplots (Warnes et al., 2009), pgirmess (Giraudoux, 2013), reshape (Wickham, 2018), tidyr (Wickham & Henry, 2017), and RColorBrewer (Neuwirth, 2011) A reproducible record of all statistical analyses is available on GitHub (https://github.com/rerodriguezmtz/ElementsSar). This includes all underlying data and R code for all analyses.

Results

The most frequent elements in sargasso tissues, detected in 100% of the samples, were As, Ca, Cl, K, Mn, P, Rb, S, Si, Sr, Th, and U. They were followed in frequency by Mg (92.1% of samples) and Al (58.7% of samples) (Table 2). Other elements were found in fewer samples and they had median concentrations below the LOD: V (28.6% of samples), Zn (12.7% of samples), and Cu, Fe, Mo and Pb, present in 7.9% of samples (Table 2). Ba, Cd, Co, Cr, Ni, Ti, Y, and Zr remained below the LOD in all the samples (See Table S1 for LOD values). Some elements showed more than 5-fold difference between their minimal and maximal concentrations (ppm DW). For example, Cl showed 71.1-fold difference, K exhibited 23.1-fold difference, As had 7.2-fold difference, Si showed 6.5-fold difference and Ca exhibited 5.7-fold difference between their minimum and maximum values (Table 2). Concentrations of P, S and Sr showed the least inter-site variability and the concentrations of Al, As, Cl and K showed the most inter-site variability (Fig. 2).

Figure 2 Spatial variability in element concentrations.

Concentration of fourteen most frequent elements (ppm algal DW) in tissues of sargasso collected at eight sites along the Mexican Caribbean coast in 2018–2019. Note differences in scale of the Y-axis. Each dot corresponds to the median of the five XRF readings per sample. Color of the dot represents the sargasso species/morphotype. The horizontal black lines correspond to the median for each site. The dotted blue line corresponds to the limit of detection of the XRF equipment. A, Aluminum; B, Arsenic; C, Calcium; D, Chlorine; E, Potassium; F, Magnesium; G, Manganese; H, Phosphorus; I, Rubidium; J, Sulphur; K, Silicon; L, Strontium; M, Thorium; N, Uranium. Figure 1 and Table 1 have the site and sample details.

Among the potentially toxic elements, only As (median contents of 24–172 ppm DW) and Mn (median contents of 40–139 ppm DW) were present in all the samples (Table 2). Of all samples, 86% presented As concentrations above the maximum allowable concentration for seaweeds to be used as animal fooder under European regulations (40 ppm DW; EU, 2019), and 100% of the samples were above the maximum allowable concentration for agricultural soils in Mexico (22 ppm DW; NOM-147-SEMARNAT-SSA1-2004). Approximately 5% of our samples showed Cu concentrations above maximum tolerable level of dietary minerals for sheep (25 ppm DW) and cattle (100 ppm DW) (McDowell, 1992). Other potentially toxic elements (e.g., Mo, Pb and Zn) were detected in only 8–13% of the samples and they had median concentrations below the toxic limits for agricultural soils (see Table 2 and Table S2).

Concentrations of As, Ca, Cl, K, Mn, Rb and Si varied significantly among sargasso species/morphotypes (Fig. 2, Kruskal–Wallis test, p < 0.05; Table 3). As, Cl, K and Rb were significantly higher in Sargassum natans VIII compared to S. natans I. The concentrations of Ca and Si were significantly lower in S. natans VIII than in S. fluitans III and S. natans I. Similarly, the concentration of Mn was higher in S. natans I compared to S. fluitans III and S. natans VIII (Table 3). Contents of Al, Mg, P, S, Sr, Th and U did not vary significantly among species and morphotypes (KW, p > 0.05; Table 3). We did not compare the concentrations of Cu, Fe, Mo, Pb and Zn statistically among the species/morphotypes as their medians remained <LOD.

Table 3 Elements concentrations in sargasso morphotypes.

Median and range (in parenthesis) of elements (ppm DW) in three sargasso species/morphotypes collected from eight localities along the Mexican Caribbean coast in 2018–2019. P values show summary of statistical analyses using Kruskal–Wallis H test (bold if significant) and the last column shows results of multiple comparison test.

Element	a) S. fluitansIII
( n = 24)	b) S. natansI
( n = 24)	c) S. natansVIII
( n = 11)	P	Multiple comparison test	
Al	221
(<LOD-392)	198
(<LOD-500)	<LOD
(<LOD-327)	0.7341		
As	59
(34–172)	55
(32–172)	123
(24–145)	0.0213	c >b	
Ca	76,727
(46,599–115,260)	81,965
(37,260–136,146)	43,289
(23,723–75,849)	0.0013	(a = b) >c	
Cl	21,487
(1,831–46,502)	10,122
(747–46,485)	32,086
(2,279–53,101)	0.0111	c >b	
K	19,466
(3,620–44,280)	14,309
(1,990–39,642)	32,900
(4,902–46,002)	0.0121	c >b	
Mg	6,376
(2,062–12,325)	6,385
(2,062–12,949)	6,883
(2,062–13,662)	0.5990		
Mn	70
(51–112)	89
(52–139)	56
(40–135)	0.0019	b >(a = c)	
P	336
(229–401)	328
(262–394)	300
(228–350)	0.0590		
Rb	60
(32–102)	51
(30–143)	67
(48–120)	0.0071	c >b	
S	14,341
(11,328–24,773)	12,776
(9,462–21,170)	16,231
(12,449–19,500)	0.1370		
Si	1,861
(927–2,877)	2,095
(696–2,564)	1,049
(447–2,135)	0.0009	(a = b) >c	
Sr	1,934
(1,641–2,395)	1,876
(1,605–2,564)	1,793
(1,633–2,362)	0.4375		
Th	10
(5–17)	8
(6–23)	9
(8–20)	0.3021		
U	22
(11–48)	23
(12–47)	27
(16–45)	0.2321		
Notes.

LOD, limit of detection.

The concentrations of fourteen different elements (i.e., Al, As, Ca, Cl, K, Mg, Mn, P, Rb, S, Si, Sr, Th and U) in sargasso collected at Puerto Morelos in seven different sampling periods, from August 2018 to April 2019, showed considerable variability (Fig. 3). This inconsistent pattern indicates absence of any seasonal tendency in the elemental concentrations.

Figure 3 Temporal variability in element concentrations.

Variability in concentration of fourteen different elements (ppm DW) in sargasso collected at Puerto Morelos between August 2018 and April 2019. Z-score transformations were applied to values of each element across all the sampling periods and their intensities above and below the mean are represented on the heatmap by red and yellow colors, respectively, as shown on the color key bar.

Discussion

The sargasso tissues from the Mexican Caribbean had more As, Cu and Mn and less Cd, Cr, Pb and Zn compared to the chemical compositions of the algae biomass from Nigeria, Ghana and Dominican Republic (Table 4). Most striking was the high variability of element concentrations detected both in space (different sites along the coast) and time (different sampling months). This variability is likely partially due to the pelagic nature of the sargasso, as a result of increased uptake when exposed to areas rich in metals. It is unlikely that heavy metals were absorbed in near-shore waters of the Mexican Caribbean because this area lacks these elements in high concentrations, due to absence of major industrial, mining or heavy agricultural activities in the region. In addition, the absorption of metals by Sargassum thunbergii under experimental conditions was only clearly noticeable after ≥3 d exposure (Wu et al., 2010), whereas the residence time of sargasso in near-shore Mexican waters is usually in the order of hours when it is transported from the Yucatan Current towards the shore. Thus, the sargasso tissues likely acquired the heavy and trace elements before entering the Mexican coastal waters. Different contaminants are released into the ocean, some as point sources and others more continuous, in different parts across the North Equatorial Recirculation Region of the Atlantic Ocean (NERR) and the Wider Caribbean Region (as a result of long-range transport). Fernandez, Singh & Jaffé (2007) recognized the discharge of sewage, mineral extracts, fertilizer and pesticide used in the agricultural sector as the principal pollution sources. The pelagic masses of sargasso might have been exposed to these contaminants depending on its trajectory in the ocean. The metal sequestration also involves complex mechanisms of ion exchange, chelation, adsorption, and ion entrapment in polysaccharide networks of the algae (Volesky & Holan, 1995). This ion entrapment, in turn, depends on the affinity of some divalent metals to alginates (Haug, 1961), and pH of the seawater also influences absorption of metals (Davis, Volesky & Vieira, 2000). Alginates are often characterized by the proportion of mannuronic (M) and guluronic (G) acids present in the polymer (M:G ratio), which may vary among and within species. For example, Mn concentration was higher in S. natans I, whereas Ca and Si concentrations were higher in S. fluitans III and S. natans I, and the concentrations of As, Cl, K and Rb were higher in S. natans VIII than in S. natans I. Variations in the metal concentrations among the sargasso species and morphological forms may be explained by different concentrations in their tissues, but also by differences in calcifying epifauna, such as bryozoans, tube polychaeta, and crustose coralline algae (Weis, 1968; Huffard et al., 2014). Large differences in concentrations of Si (447–2,922 ppm DW) could be explained by different abundance of diatoms and silicoflagellates present in the samples (Takahashi & Blackwelder, 1992).

Table 4 Element concentrations in different studies.

Comparison of element concentration in sargasso from the Mexican Caribbean coast and other studies in different parts of the world.

Element	Site (Year)				
	Nigeriaa (2012)	Dominican Republicb (2015)	Ghanac (2015)	Mexican Caribbeand
(2018–2019)	
Al		303–4,188		<140–517	
As		14–42	13–54	24–172	
Ba		7–17		<36	
Ca		96,901–133,400		23,723–136,146	
Cd		0.1–0.3	78–119	<2	
Cl			61–1353	747–53,101	
Co		0.4–1		<11	
Cr		2–56		<8	
Cu		2–12	24–36	<6–540	
Fe	8,700 ± 280	20–655	1,209–5,910	<3–11	
K	28,000 ± 740	2,208–33,602		1,990–46,002	
Mg	42,750 ± 3,500	10,211–18,241		<2915–13,662	
Mn		16–32		40–139	
Mo		0.6–3		<1–7	
Ni		10–33		<10	
P	96,500 ± 21,200	761–1,145		228–401	
Pb		1–2	105–335	<2–3	
Rb		0.3–10		30–143	
Si		23,883–55,776		447–2,922	
Sr		1,162–1,437		1,605–2,564	
Th		0.04–0.4		5–23	
Ti		37–92		<29	
U		0.2–0.7		11–48	
V		1–3		<3–13	
Y	40 ± 0.0	0.1–0.8		<1	
Zn	50 ± 0.0	13–21	16–100	<5–17	
Zr		8–34		<2	
Notes.

a Oyesiku & Egunyomi, 2014 (mean and SD).

b Fernández et al., 2017 (range).

c Addico & De Graft-Johnson, 2016 (range).

d This study (range).

Sargasso samples from the Mexican Caribbean coast contained essential macro-elements for plants, like Ca (23,723–136,146 ppm DW), K (1,990–46,002 ppm DW), Mg (<2,915–13,662 ppm DW), P (228–401 ppm DW) and S (9,462–24,773 ppm DW), in addition to various micro-elements. Similar properties have been found in other Sargassum spp., making them adequate as complementary fertilizers as they enhance growth, seed germination and photosynthesis of crop plants on mineral-depleted soils (Sathya et al., 2010; Kumari, Kaur & Bhatnagar, 2013; El-Din, 2015). Some micro-elements found in sargasso from Mexico, such as Cu, Mn, Mo and Zn, are micronutrients in low concentrations, but they are potentially toxic when present in high concentrations. In this study, we detected the presence of Cu (<8–540 ppm DW) and Mo (<1–7 ppm DW) in 7.9% of the samples, Zn (<2–17 ppm DW) in 12.7% of the samples and Mn (40-139 ppm DW) in all the samples. Cu concentrations exceeded safely limits recommended for agricultural soils by several countries in 5% of the samples (see Table S2). Similarly, about 8% of our samples contained Mo concentrations above the maximum level established for agricultural soils by Canada (i.e., 2 ppm DW), but these were below the limits established by Austria and Poland (i.e., 10 ppm DW). Mn content was above 100 ppm DW in 22% of the samples, considered toxic for some plant species, but acceptable for others that can tolerate Mn up to 5,000 ppm DW (Howe, Malcolm & Dobson, 2004). Pb (<2–3 ppm DW) could be detected only in 7.9% of the samples, due to the limitation related to LOD of XRF analysis, and its concentration always remained below the toxic levels. Arsenic is of concern for the usages of sargasso as complementary fertilizer for crop plants. Limits of total As allowed for agricultural soils are between 15–50 ppm DW depending on the country (Table S2) (Belmonte-Serrato et al., 2010), thus, continuous application of sargasso (with total As between 24–172 ppm DW) may cause accumulation of As in the soils above allowable levels. High concentrations of As in soil may be toxic for the plants themselves, as it interferes with photosynthesis and other metabolic processes (Påhlsson, 1989; Ruiz Huerta & Armienta Hernández, 2012).

Sargasso could also be considered as animal fodder due to the presence of micro- and macro-elements, in addition to proteins, fibers and other components (Marín et al., 2009; Carrillo et al., 2012). However, approximately 86% of the samples had total As concentrations above the maximum level (40 ppm DW) allowable in Europe for animal feed materials derived from seaweed (EU, European Union). The toxicity of As depends on its chemical form, with inorganic As (trivalent state As III and pentavalent state As V) considered toxic (e.g., Yuan et al., 2007, Circuncisão et al., 2018), thus, even if total As concentrations are below 40 ppm DW, it is recommendable to carry out As speciation studies before using sargasso as animal fodder.

The (occasional) high contents of potentially toxic metals in sargasso is also a serious threat for the environment. The Mexican Caribbean coast has already received millions of tons of algae since late 2014. This accumulation over time, in addition to eutrophication and organic matter accumulation (Carruthers, van Tussenbroek & Dennison, 2005; Hernández-Terrones et al., 2011; Baker, Rodríguez-Martínez & Fogel, 2013; van Tussenbroek et al., 2017), is also a potential source of metal contamination for this region, even though levels of some potentially toxic elements like Cu, Mo, Zn, Mn and Pb were low. The sargasso removed from Mexican Caribbean beaches is presently deposited at abandoned limestone quarries, near the coast, without any treatment. The Yucatan Peninsula has a highly porous karst aquifer that is the only source of freshwater in the region. The pollutants from near surface deposits can easily infiltrate into the aquifer causing accumulation of As and other potentially toxic metals in the groundwater. Considering that water from the aquifer flows into the ocean through underground rivers, all these metals and excessive nutrients will eventually reach the marine environment (Carruthers, van Tussenbroek & Dennison, 2005; Metcalfe et al., 2011; Baker, Rodríguez-Martínez & Fogel, 2013). Prevention and mitigation measures are urgently needed to ensure that the massive influx of sargasso does not harm the coastal ecosystems and the tourism-based economy of countries located in the vicinity of the Great Atlantic Sargassum belt, including the Mexican Caribbean. The analyses of different specimens collected over longer periods and from different locations is required to obtain reliable information about metal contents in tissues.

Conclusion

In countries affected by the Great Atlantic Sargassum belt, the accumulation of decomposing sargasso on shores has harmed the coastal ecosystems, tourism-based economy and general human well-being. The Mexican Caribbean coast has received millions of tons of sargasso since late 2014, and our study concludes that the massive influx might contribute with potentially toxic elements to the coastal ecosystems, including the aquifer. We observed relatively higher values of As, Cu and Mn and lower values of Cd, Cr and Pb compared to similar studies in countries affected by the Sargassum belt. Cu, Mo, Zn, Mn and Pb were present in lower contents but their accumulation over time might be a potential source of contamination in this region. Total arsenic in most samples exceeded the limit established for usage as animal fodder in Europe and for agricultural soil in several countries. Further studies on As speciation are required before using sargasso in food industries to determine if it complies with guidelines of international institutions and organizations (i.e., FAO, WHO). Chemical analysis should also be conducted using other methodologies such as an ICP-MS, with better limit of detection, before evaluating sargasso usages in food, pharmaceutical and agricultural industries. Governments and industries have the financial strengths, as well as the moral and legal responsibilities, to carry out regular analyses of specimens collected over long periods and from different locations required for obtaining reliable information about metal contents in the tissues of sargasso due to its unpredictable variability.

Supplemental Information

Supplemental Information 1 Limits of Detection (LOD) of the analyzed elements in Niton FXL energy dispersive XRF (ppm = mg/kg) and Toxic metals and trace elements maximum levels permitted by different countries in agricultural soils (ppm = mg kg−1). nr: no reported

Click here for additional data file.

Special thanks to Elisa Vera Vázquez, for collecting sargasso samples from Puerto Morelos and Manta México Caribe A.C. for providing samples from Blue Waters and Isla Contoy. We also want to thank CEMIE-Oceano for the samples at Cozumel. Gabriela González López provided logistic help during fieldwork and in laboratory and Irma Gabriela Vargas-Martinez helped in XRF analysis. The authors claim no conflict of interest with this work.

Additional Information and Declarations

Competing Interests

Author Contributions

Field Study Permissions

Data Availability

The authors declare there are no competing interests.

Rosa E. Rodríguez-Martínez conceived and designed the experiments, analyzed the data, prepared figures and/or tables, authored or reviewed drafts of the paper, and approved the final draft.

Priyadarsi D. Roy, Nuria Torrescano-Valle and Nancy Cabanillas-Terán conceived and designed the experiments, performed the experiments, authored or reviewed drafts of the paper, and approved the final draft.

Silvia Carrillo-Domínguez, Ligia Collado-Vides, Marta García-Sánchez and Brigitta I. van Tussenbroek conceived and designed the experiments, authored or reviewed drafts of the paper, and approved the final draft.

The following information was supplied relating to field study approvals (i.e., approving body and any reference numbers):

All surveys were conducted under permit PPD/DGOPA-116/14granted by SAGARPA (Agriculture, Natural Resources and Fisheries Secretariat) to Brigitta I. van Tussenbroek.

The following information was supplied regarding data availability:

Data is available at https://github.com/rerodriguezmtz/ElementsSar.

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
