# Peer review of "Element concentrations in pelagic Sargassum along the Mexican Caribbean coast in 2018-2019"

_PeerJ, doi:10.7717/peerj.8667_

## Round 0.1 · original submission · Minor Revisions

Please revise your manuscript according to reviewer comments. Once you have completed the revisions, I would encourage you to read the manuscript carefully for grammar.

Reviewer 1 ·

Basic reporting

This submission is topical and will be of interest to all those researching and effected by Golden-tide problem in the Gulf of Mexico and the Caribbean.

It is generally well written, although there is some awkward phraseology, such as line 197.

The scientific question and experimental set up seem sound with appropriate statistical methods. The figures are clear and have sufficient captions for them to be read stand-alone.

The primary method is XRF, which, although novel, is not the standard method of 'heavy metal' analysis.; more discussion on the strengths and weaknesses of this method relative to the more usual procedures such as ICP may be of value.

Experimental design

The scientific question and experimental set up seem sound with appropriate statistical methods. The figures are clear and have sufficient captions for them to be read stand-alone.

Validity of the findings

The primary method is XRF, which, although novel, is not the standard method of 'heavy metal' analysis.; more discussion on the strengths and weaknesses of this method relative to the more usual procedures such as ICP may be of value.

Additional comments

This submission is topical and will be of interest to all those researching and effected by Golden-tide problem in the Gulf of Mexico and the Caribbean.

It is generally well written, although there is some awkward phraseology, such as line 197.

The scientific question and experimental set up seem sound with appropriate statistical methods. The figures are clear and have sufficient captions for them to be read stand-alone.

The primary method is XRF, which, although novel, is not the standard method of 'heavy metal' analysis.; more discussion on the strengths and weaknesses of this method relative to the more usual procedures such as ICP may be of value.

Reviewer 2 ·

Basic reporting

no comment

Experimental design

Because the temporal variability was analyzed, the collection date of each sample should be given in Table1. I am also concerned with the very few samples in some locations. As there were so many floating Sargassum in the sea, why only one sample was collected in some places. The replication for these locations was not enough. At least the authors should describe the reason for the limited number of samples.

Validity of the findings

no comment

Additional comments

I do not agree on one point in the first paragraph of Discussion.
The authors wrote the heavy metals were unlikely adsorbed nearshore. But as I know, the adsorption process is usually very fast, so it cannot exclude this possibility.

In the method section:

Did you handle the samples before drying them? As you have discussed, epiphytes may adsorb or absorb metals. Did you remove the epiphytes before the analysis?

I am concerned with the limited sampling size in some locations. If this issue can be addressed reasonably, I recommend the acception of this MS as basically it is well organized and well written.

---

## Round 0.2 · accepted · Accept

Thank you for your efforts to revise your manuscript based on reviewer comments.

Reviewer 1 ·

Basic reporting

No further comment

Experimental design

No additional comments

Validity of the findings

No further comment

Additional comments

The authors have fully answered my queries and substantially improved grammar and phraseology. However, please check Table 1 in the supplementary materials there are citations with superscripts, but no superscript entries in the table.